# Study on the Mechanism of Motion Interaction between Soil and a Bionic Hole-Forming Device

Long Wang [1,2], Jianfei Xing [1,*], Xiaowei He [1], Xin Li [3], Wensong Guo [1], Xufeng Wang [1] and Shulin Hou [2]

[1]  College of Mechanical and Electronic Engineering, Tarim University, Alar 843300, China;
    120140002@taru.edu.cn (L.W.); 120140001@taru.edu.cn (X.H.); 120120004@taru.edu.cn (W.G.);
    wxf@taru.edu.cn (X.W.)
[2]  College of Engineering, China Agricultural University, Beijing 100083, China; h01520@cau.edu.cn
[3]  School of Mechanical Engineering, Hunan University of Technology, Zhuzhou 412007, China;
    12177@hut.edu.cn
[*]  Correspondence: 120200012@taru.edu.cn

**Abstract:** Due to the lack of water resources, the main agricultural planting method used in the northwest region of China is plastic film mulching, with precision hole sowing performed on the film after mulching. However, conventional hole-forming devices damage the compactness of the soil hole while moving on the plastic film, causing seed misplacement. Therefore, this study designed a bionic hole-forming device based on the oriental mole cricket. In order to explore the interaction between the hole-forming device and the soil, a typical soil discrete element particle model was established, and its contact parameters were calibrated. An experiment was conducted to compare the performance of the bionic hole-forming device with a conventional device using discrete element method and multi-body dynamics (DEM-MBD) coupled simulations. The results revealed that the bionic hole-forming device caused less soil disturbance during the hole-forming process and could reduce the sowing operation resistance compared to the traditional device. Compared to traditional square and cone-shaped hole-forming devices, the soil resistance of the bionic hole-forming device was the smallest, at 7.51 N. This work provides a reference for the optimization of hole-forming devices for plastic film sowing.

**Keywords:** discrete element method; bionic technology; soil; hole-forming device

## 1. Introduction

Due to its unique geographical location and natural environment, the northwest region of China has emerged as a major cotton planting base for the country [1]. The primary method of seeding cotton is through hole seeding on plastic film. However, existing hole-forming devices used in this process create large holes in the plastic film and seriously disturb the soil [2]. Therefore, there is an urgent need to develop hole-forming machinery suitable for the soil conditions of the farmland in Northwest China to reduce soil disturbance and minimize operational resistance. Such equipment would greatly contribute to the sustainable development of the cotton industry.

In order to successfully design soil-engaging components for agricultural machinery, the interaction mechanism between the soil-engaging components and the soil must be clarified [3]. The use of soil trenches and field tests can only provide a macroscopic analysis of soil disturbance and is unable to investigate the soil's movement patterns and force characteristics at the microscopic scale [4]. Numerous studies have demonstrated the applicability of the discrete element method to simulate and analyze the interaction between soil-engaging components and soil and to optimize the structural parameters of the machinery [5–8]. However, the type and properties of the soil directly affect the interaction effect between the agricultural machinery components and soil, and the accuracy of the soil model and contact parameters have a significant impact on the simulation results [9,10].

Currently, much research, both domestic and foreign, has been conducted on soil parameter calibration and soil–machinery interactions based on the discrete element method [11–14]. Aikins et al. developed a high-viscosity soil model by combining the hysteretic spring and linear cohesion models. The static and rolling coefficients of friction were calibrated, and the accuracy of the model was verified through trenching experiments [15]. In a study on the interaction between soil and a plate-type plow using the discrete element method, Ucgul et al. compared the working traction force and the profile of the plowed furrow through simulations and field experiments. The authors revealed the ability of the discrete element method to accurately simulate the actual working processes [16]. Milkevych et al. established a discrete element model for the interaction between the soil and soil-engaging components for weed control operations. The authors analyzed the impact of weed control operations on soil displacement and validated the effectiveness of the discrete element simulation through the consistent results obtained from both simulations and field experiments [17]. Shaikh et al. established a soil–grouser interaction model and simulated the interaction of a single grouser shoe with clay loam terrain at a varying moisture content by the DEM with Hertz–Mindlin contact [18]. Xiang et al. calibrated the contact parameters of a discrete element model for cohesive soil based on soil pile-up experiments and validated the effectiveness of the calibrated parameters through excavation tests [19]. Based on the hysteretic spring and linear cohesion models, Ma Shuai et al. calibrated the contact model parameters between frozen soil particles and validated the accuracy of the model parameters through a comparative analysis of trench and simulation experiments [20]. Zhang et al. constructed a discrete element simulation model of the contact between sticky black soil and agricultural implements based on the Hertz–Mindlin and JKR cohesion models. By combining simulations and trench experiments, the authors evaluated the micro-disturbance mechanism and macro-disturbance state of soil caused by plowing and shoveling [21].

In summary, establishing an accurate soil discrete element particle model is crucial in order to accurately investigate the interaction between agricultural machinery components and soil [22–24]. In this study, physical soil characteristic parameters, including moisture content, density, hardness, particle size distribution, and the angle of repose were measured in typical cotton fields within the northwest region of China during the suitable seeding period. A soil discrete element particle model was constructed, and the contact parameters of the model were calibrated based on experiments and discrete element simulations. A bionic hole-forming device for cotton was designed and compared with conventional square and cone-shaped hole-forming devices through simulation experiments. The purpose of this study was to: (1) perform discrete element method and multi-body dynamics (DEM-MBD) modeling to analyze hole-forming processes and performance under various hole-forming devices, and (2) provide optimization approaches for the improvement of hole-forming devices.

## 2. Materials and Methods

### 2.1. Experiment Material

Soil samples were collected from a cotton test field located in Tamutugrak Township, Xinhe County, Northwest China, with the geographical coordinates 82°25′26″ E and 41°16′43″ N and an altitude of 952 m. In order to determine the model parameters, the samples were collected using the five-point sampling method from the plowed and harrowed cotton field soil during the suitable sowing period (March to April). The sampling depth was set as 10 cm, as the cotton seeder had an impact within 10 cm of the soil layer.

### 2.2. Soil Particle Size Distribution

The soil sieving method was employed to determine the particle size distribution of the soil in the cotton fields during the suitable sowing period. A high-frequency vibrating screen (GZS-1, TuoZhan Instrument Equipment Co., Ltd., Taizhou, China) with a 500 times/min vibration frequency and 1.5 mm vibration amplitude was used for the mea-

surements along with different sieve meshes. During the experiment, sieve meshes with pore diameters of 2.0 mm, 1.0 mm, and 0.075 mm were placed on the high-frequency vibrating screen [9]. A certain amount of soil was weighed and placed on the top sieve, the top cover was closed, and the sieve meshes were fixed prior to turning on the high-frequency vibrating screen. The vibrating screen was then run for 10 min and subsequently turned off. The sieve meshes were then removed (from the largest to smallest pore diameter), and the soil samples in each sieve were taken out. The soil samples adhering to the sieve meshes were cleaned with a soft brush, and each soil sample was weighed. The experiment was repeated three times. Table 1 reports the results.

**Table 1.** Particle size distribution of soil.

| Samples | Particle Size Ratio of Soil (%) | | |
|---|---|---|---|
| | Gravel (>1 mm) | Sand (0.075–1 mm) | Silt (<0.075 mm) |
| 1 | 34.12 | 52.26 | 13.62 |
| 2 | 35.24 | 51.67 | 13.09 |
| 3 | 34.98 | 52.43 | 12.59 |
| Average | 34.78 | 52.12 | 13.10 |

The proportion of gravel (particle size > 1 mm), sand (particle size 0.075–1 mm), and silt (particle size < 0.075 mm) in the soil of the experimental farmland was 34.78%, 52.12%, and 13.1%, respectively (Table 1).

### 2.3. Soil Moisture Content and Density

The soil moisture content was measured by the constant pressure and temperature drying method using an electric hot air oven (GZX-9140MBE, Bosen Instrument Co., Ltd., Shanghai, China); electronic balance (CL-T500, Chenlong Hengxin Trading Co., Ltd., Beijing, China, 0.001 g accuracy); and vacuum sample drying box. During the experiment, the aluminum boxes of each sample were labeled and weighed. The soil samples were then placed in the aluminum boxes, weighed, and placed in the drying oven with the temperature set at 105 °C. Every two hours, the samples were taken out for weighing until the weight became constant, and the final weight was recorded. The soil moisture content of the farmland was calculated according to Formula (1):

$$M = \frac{m_0 - m_1}{m_0} \times 100\%,\tag{1}$$

where $M$ is the moisture content of the soil (%), $m_0$ is the mass of the soil before drying (g), and $m_1$ is the mass of the soil after drying (g).

The soil moisture content test was repeated three times. The results determined that the average moisture content of the soil layer at a 10 cm depth equaled 14.46%.

The soil density of the cotton field during the suitable seeding period was measured with the ring knife method. A soil sampling ring knife (ZLT, Zhonglutong, testing instrument Co., Ltd., Suzhou, China) with a 100 cm$^3$ volume and electronic balance (CL-T500, Chenlong Hengxin Trading Co., Ltd., Beijing, China, 0.001 g accuracy) were used for the measurements. The mass of the aluminum sample box was first weighed using the electronic balance. A 100 cm$^3$ volume of soil was collected using the soil sampling ring knife within the 10 cm soil layer and placed in the aluminum sample box for weighing. The soil density was then calculated according to Formula (2):

$$\rho_s = \frac{m_s}{V_s},\tag{2}$$

where $\rho_s$ is the soil density (g/cm$^3$), $m_s$ is the weight of the undisturbed soil (g), and $V_s$ is the volume of the undisturbed soil (100 cm$^3$).

The soil density measurements were repeated three times. The average soil density in the 10 cm soil layer during the suitable seeding period of the cotton field was determined as 1.37 g/cm$^3$.

### 2.4. Soil Firmness

A soil firmness tester (LD-TJ, Shandong Lai Ende Intelligent Technology Co., Ltd., 0.1% measurement accuracy) was used to measure the soil firmness of the cotton field during the suitable seeding period. Figure 1 depicts the measurement process. The five-point sampling method was used for the measurements. The testing probe was placed vertically on each of the five testing points. The operation panel data were cleared by selecting a "zero clearing" operation. The probe was pressed into the soil at a depth of 10 cm, and the experimental data were recorded. The average soil firmness was calculated based on the data from the five testing points.

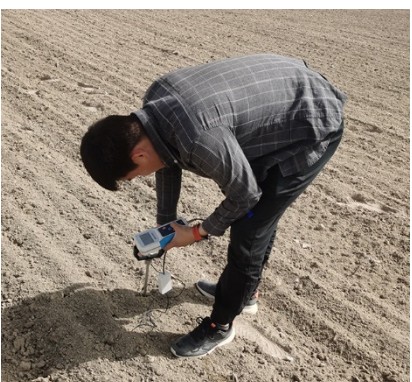

**Figure 1.** Firmness determination of soil.

The soil firmness test was repeated three times, and the firmness of the soil in the cotton field during the suitable seeding period was determined as 1547.7 ± 89.6 Pa.

### 2.5. Soil Stacking Angle

Soil pile experiments were performed to measure the soil pile angle of the cotton field during the suitable seeding period. A funnel (14.0 cm top diameter, 2.6 cm bottom diameter, and 12.5 cm height); stand; material dropping plate; ruler; and inclination angle meter (ZLT, Zhonglutong Testing Instrument Co., Ltd., Suzhou, China, accuracy of 0.2°) were used for the measurements. The height between the funnel mouth and bottom plate was determined by adjusting the position of the bracket. During the experiment, the distance between the funnel mouth and bottom plate was first adjusted to 15 cm, and the soil was evenly poured into the funnel. The soil pile angle was measured with a ruler and an inclination angle meter. Figure 2 presents the measurement process. In order to reduce experimental errors, four test directions were taken for each soil pile angle test, and the average value of the test results was taken as the soil pile angle of that test. The soil pile angle test was repeated three times. Table 2 reports the results.

**Table 2.** Soil stacking angle measurement results.

| No. | Stacking Angle of Soil (°) | | | | | |
| --- | --- | --- | --- | --- | --- | --- |
| | Orientation 1 | Orientation 2 | Orientation 3 | Orientation 4 | Average | Total Average |
| 1 | 39.30 | 38.25 | 37.60 | 40.55 | 38.93 | |
| 2 | 39.25 | 42.60 | 40.50 | 37.45 | 39.95 | 39.34 |
| 3 | 38.60 | 39.45 | 41.85 | 36.70 | 39.15 | |

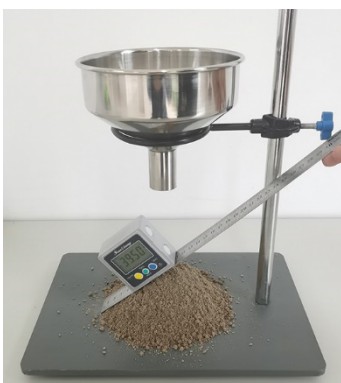

**Figure 2.** Stacking angle determination of soil.

The minimum and maximum repose static angles of the cotton field soil during the sowing period were determined as 37.45° and 42.60°, respectively, and the soil accumulation as 39.34 ± 1.71°.

*2.6. Discrete Element Contact Model of Soil*

Soil particles adhere to each other due to the influence of water and other chemical substances, resulting in the phenomenon of aggregation [25]. The Hertz–Mindlin (no-slip) contact model cannot accurately simulate this mechanical phenomenon of soil [26]. However, combining the cohesive force of the Johnson–Kendall–Roberts (JKR) model with the Hertz–Mindlin model can accurately represent the influence of various cohesive effects [27]. By fully considering the influence of the Van der Waals forces among particles within the contact range, it is possible to simulate adhesive and cohesive systems such as fine powders, dry powders, and wet materials [28]. Therefore, the combined Hertz–Mindlin and JKR contact model is generally adopted to simulate the mechanical behavior of soil. In this model, the JKR normal elastic contact force is based on the Johnson–Kendall–Roberts theory, which characterizes the overlap between particles, interaction parameters, and surface energy. The other forces are calculated in the same way as the Hertz–Mindlin (no-slip) contact model. The JKR normal elastic force is described in Formula (3):

$$F_{\mathrm{JKR}} = -4\sqrt{\pi\gamma E_{\mathrm{eq}}}\alpha^{\frac{3}{2}} + \frac{4E_{\mathrm{eq}}}{3R_{\mathrm{eq}}}\alpha^3, \tag{3}$$

$$\delta = \frac{\alpha^2}{R_{\mathrm{eq}}} - \sqrt{\frac{4\pi\gamma\alpha}{E_{\mathrm{eq}}}}, \tag{4}$$

where $F_{\mathrm{JKR}}$ is the JKR normal elastic force (N), $\alpha$ is the radius of the contact circle of the two contacting particles (m), $\delta$ is the normal distance between the particles (m), and $\gamma$ is the surface energy (J/m$^2$).

When there is a certain gap between two particles, this contact model can also be used to calculate the cohesive force between the particles. The maximum distance between the two particles with a non-zero cohesive force can be calculated using Formula (5):

$$\delta_{\mathrm{c}} = \frac{\alpha_{\mathrm{c}}^2}{R_{\mathrm{eq}}} - \sqrt{\frac{4\pi\gamma\alpha_{\mathrm{c}}}{E_{\mathrm{eq}}}}, \tag{5}$$

$$\alpha_{\mathrm{c}} = \left[\frac{9\pi\gamma R_{\mathrm{eq}}}{2E_{\mathrm{eq}}}\left(\frac{3}{4} - \frac{1}{\sqrt{2}}\right)\right]^{\frac{1}{3}}, \tag{6}$$

where $\delta_{\mathrm{c}}$ is the maximum distance between particles with non-zero cohesive force (m), and $\alpha_{\mathrm{c}}$ is the maximum contact circle radius between two contacting particles (m).

When the normal gap between the particles is greater than $\delta_c$, the cohesive force between the particles is zero. If the particles are not in contact with each other and the normal gap between them is less than $\delta_c$, the maximum value of the cohesive force can be calculated using Formula (7):

$$F_{\text{pullout}} = -\frac{3}{2}\pi\gamma R_{\text{eq}}, \tag{7}$$

where $F_{\text{pullout}}$ is the maximum value of the cohesive force (N).

The combined Hertz–Mindlin and JKR contact model can provide a frictional force for a large cohesive force component in the contact normal and can accurately simulate the mechanical behavior of soil. As the direct target of the hole-forming process was the soil in a typical cotton field in Northwest China during the suitable sowing period, the combined Hertz–Mindlin and JKR model was selected as the contact model between soil particles in the simulation process.

### 2.7. Stacking Angle Test

In this study, the contact parameters between soil and soil and between soil and steel were calibrated using simulation experiments. The soil pile angle was simulated and measured. During the simulation experiment, the funnel size and distance between the funnel outlet and steel plate were consistent with the soil pile angle measurement experiment in soil stacking angle test. According to the soil particle size distribution experiment, the soil particle size was mainly concentrated around 1 mm, and the soil particle radius was set as 1 mm in the simulation. The soil particles were randomly distributed within 0.8–1.2 times the standard volume and generated in a planar manner on the funnel. The soil particles fell through the funnel in a free fall, and the total mass of soil particle production was 200 g. The simulation ended when the soil particles were piled up and stabilized on the flat plate. The Protractor module in EDEM was used to measure the simulation static angle of the soil. Figure 3 depicts the measurement process.

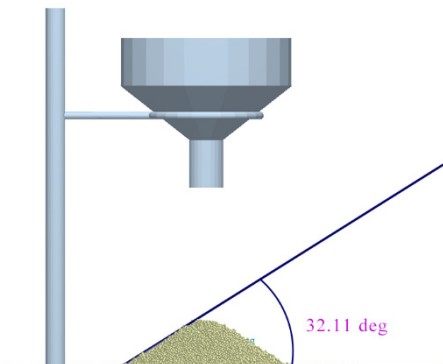

**Figure 3.** Determination of simulated soil stacking angle.

We determined the Plackett–Burman screening experimental design using Design-Expert 12.0 to screen the contact parameters that had a significant impact on the soil stacking angle. The soil stacking angle was set as the response value. There were a total of seven parameters to be calibrated in the simulation experiment, labeled as $X_1$ to $X_7$, and four virtual parameters labeled as $X_8$ to $X_{11}$. Based on a large number of preliminary trials and relevant literature, the following parameter values were set: the Poisson's ratio of the cotton field soil was 0.36; the shear modulus was $1.0 \times 10^6$ Pa; the soil-to-soil collision restoration coefficient was 0.2–0.6; the soil-to-soil static friction coefficient was 0.2–0.6; the soil-to-soil dynamic friction coefficient was 0.05–0.35; the soil-to-steel collision restoration coefficient was 0.3–0.7; the soil-to-steel static friction coefficient was 0.2–0.6; the soil-to-steel dynamic friction coefficient was 0.05–0.25; and the JKR surface energy was 0.05–0.35 J/m². The Plackett–Burman experimental parameter design list is shown in Table 3.

**Table 3.** List of Plackett–Burman test parameters.

| Symbol | Parameter | Parameter Level | | |
|---|---|---|---|---|
| | | −1 | 0 | 1 |
| $X_1$ | Soil-to-soil collision restoration coefficient | 0.2 | 0.4 | 0.6 |
| $X_2$ | Soil-to-soil static friction coefficient | 0.2 | 0.4 | 0.6 |
| $X_3$ | Soil-to-soil dynamic friction coefficient | 0.05 | 0.2 | 0.35 |
| $X_4$ | Soil-to-steel collision restoration coefficient | 0.3 | 0.5 | 0.7 |
| $X_5$ | Soil-to-steel static friction coefficient | 0.2 | 0.4 | 0.6 |
| $X_6$ | Soil-to-steel dynamic friction coefficient | 0.05 | 0.15 | 0.25 |
| $X_7$ | JKR surface energy ($J/m^2$) | 0.05 | 0.25 | 0.35 |
| $X_8$–$X_{11}$ | Virtual parameters | | | |

Table 4 reports the design plan and simulation results of the Plackett–Burman screening experiment for the soil simulation parameters. The impacts of the calibrated parameters were obtained through variance analysis (Table 5). Compared with other parameters, the soil-to-soil static friction coefficient $X_2$, soil-to-soil dynamic friction coefficient $X_3$, soil-to-steel plate static friction coefficient $X_5$, and JKR surface energy $X_7$ had a significant impact on the soil simulation repose angle, while the remaining parameters had no significant effect on the soil pile angle. Therefore, in the curved surface response test, the optimization of the soil-to-soil static friction coefficient $X_2$, soil-to-soil dynamic friction coefficient $X_3$, soil-to-steel plate static friction coefficient $X_5$, and JKR surface energy $X_7$ was sufficient.

**Table 4.** Schemes and results of Plackett–Burman test.

| No. | $X_1$ | $X_2$ | $X_3$ | $X_4$ | $X_5$ | $X_6$ | $X_7$ | $X_8$ | $X_9$ | $X_{10}$ | $X_{11}$ | Stacking Angle of Soil (°) |
|---|---|---|---|---|---|---|---|---|---|---|---|---|
| 1 | 1 | 1 | −1 | 1 | 1 | 1 | −1 | −1 | −1 | 1 | −1 | 28.57 |
| 2 | −1 | 1 | 1 | −1 | 1 | 1 | 1 | −1 | −1 | −1 | 1 | 51.90 |
| 3 | 1 | −1 | 1 | 1 | −1 | 1 | 1 | 1 | −1 | −1 | −1 | 50.16 |
| 4 | −1 | 1 | −1 | 1 | 1 | −1 | 1 | 1 | 1 | −1 | −1 | 45.32 |
| 5 | −1 | −1 | 1 | −1 | 1 | 1 | −1 | 1 | 1 | 1 | −1 | 37.19 |
| 6 | −1 | −1 | −1 | 1 | −1 | 1 | 1 | −1 | 1 | 1 | 1 | 42.74 |
| 7 | 1 | −1 | −1 | −1 | 1 | −1 | 1 | 1 | −1 | 1 | 1 | 49.88 |
| 8 | 1 | 1 | −1 | −1 | −1 | 1 | −1 | 1 | 1 | −1 | 1 | 23.18 |
| 9 | 1 | 1 | 1 | −1 | −1 | −1 | 1 | −1 | 1 | 1 | −1 | 45.17 |
| 10 | −1 | 1 | 1 | 1 | −1 | −1 | −1 | 1 | −1 | 1 | 1 | 26.42 |
| 11 | 1 | −1 | 1 | 1 | 1 | −1 | −1 | −1 | 1 | −1 | 1 | 40.51 |
| 12 | −1 | −1 | −1 | −1 | −1 | −1 | −1 | −1 | −1 | −1 | −1 | 24.43 |

**Table 5.** Parameter significance analysis of Plackett–Burman test.

| Parameter | Degrees of Freedom | Mean Square Sum | *F*-Value | *p*-Value | Significance |
|---|---|---|---|---|---|
| $X_1$ | 1 | 7.47 | 6.66 | 0.0613 | |
| $X_2$ | 1 | 49.41 | 44.02 | 0.0027 | ** |
| $X_3$ | 1 | 115.51 | 102.91 | 0.0005 | ** |
| $X_4$ | 1 | 0.3234 | 0.2881 | 0.6199 | |
| $X_5$ | 1 | 141.93 | 126.45 | 0.0004 | ** |
| $X_6$ | 1 | 0.3367 | 0.3000 | 0.6130 | |
| $X_7$ | 1 | 916.48 | 816.51 | <0.0001 | ** |

Note: ** denotes an extremely significant impact from the parameter ($p < 0.01$).

Based on the results of the screening experiment, with the soil pile angle as the indicator, the significant parameters including the soil-to-soil static friction coefficient $X_2$, soil-to-soil dynamic friction coefficient $X_3$, soil-to-steel plate static friction coefficient $X_5$, and JKR surface energy $X_7$ were gradually increased according to the selected step size,

and the remaining parameters were simulated at the intermediate level. Table 6 reports the simulation test factor encoding.

**Table 6.** Coding of simulation test factors.

| Code | $X_2$ | $X_3$ | $X_5$ | $X_7$ |
|---|---|---|---|---|
| −2 | 0.2 | 0.05 | 0.2 | 0.05 |
| −1 | 0.3 | 0.125 | 0.3 | 0.125 |
| 0 | 0.4 | 0.20 | 0.4 | 0.20 |
| 1 | 0.5 | 0.275 | 0.5 | 0.275 |
| 2 | 0.6 | 0.35 | 0.6 | 0.35 |

Based on the factor encoding of the simulation test shown in Table 6, a Box–Behnken experiment of the surface response was designed. A total of 30 tests were conducted. Table 7 reports the simulation test plan and results.

**Table 7.** Box–Behnken experimental design and results.

| No. | $X_2$ | $X_3$ | $X_5$ | $X_7$ | Stacking Angle of Soil (°) |
|---|---|---|---|---|---|
| 1 | −1 | −1 | 1 | 1 | 45.09 |
| 2 | 0 | 0 | 2 | 0 | 43.30 |
| 3 | 1 | 1 | 1 | 1 | 44.73 |
| 4 | 0 | 2 | 0 | 0 | 43.75 |
| 5 | 0 | 0 | 0 | 0 | 40.99 |
| 6 | 1 | 1 | 1 | −1 | 40.47 |
| 7 | 0 | 0 | 0 | 0 | 41.91 |
| 8 | 0 | 0 | 0 | 0 | 41.88 |
| 9 | −1 | −1 | −1 | −1 | 36.09 |
| 10 | 0 | −2 | 0 | 0 | 38.60 |
| 11 | −1 | 1 | 1 | 1 | 47.94 |
| 12 | 0 | 0 | 0 | 0 | 42.41 |
| 13 | 0 | 0 | 0 | 0 | 41.66 |
| 14 | −1 | 1 | −1 | −1 | 38.59 |
| 15 | 2 | 0 | 0 | 0 | 38.00 |
| 16 | 0 | 0 | 0 | −2 | 34.61 |
| 17 | 1 | 1 | −1 | 1 | 43.64 |
| 18 | 1 | 1 | −1 | −1 | 35.24 |
| 19 | 0 | 0 | 0 | 2 | 49.71 |
| 20 | −2 | 0 | 0 | 0 | 44.52 |
| 21 | 1 | −1 | 1 | −1 | 36.48 |
| 22 | 0 | 0 | −2 | 0 | 34.04 |
| 23 | −1 | 1 | 1 | −1 | 40.18 |
| 24 | 1 | −1 | 1 | 1 | 44.57 |
| 25 | 1 | −1 | −1 | −1 | 35.17 |
| 26 | −1 | −1 | −1 | 1 | 44.56 |
| 27 | −1 | 1 | −1 | 1 | 43.04 |
| 28 | 1 | −1 | −1 | 1 | 44.04 |
| 29 | 0 | 0 | 0 | 0 | 43.89 |
| 30 | −1 | −1 | 1 | −1 | 38.05 |

*2.8. Design of Novel Bionic Hole-Forming Device*

The oriental mole cricket has an extremely strong soil digging ability. Its forelegs are nail-shaped (Figure 4). The wedge angle of its claws reduces resistance during the soil digging process [29]. In this study, the outer contour of the foreleg claws of the oriental mole cricket was employed to design a new type of bionic hole-forming device.

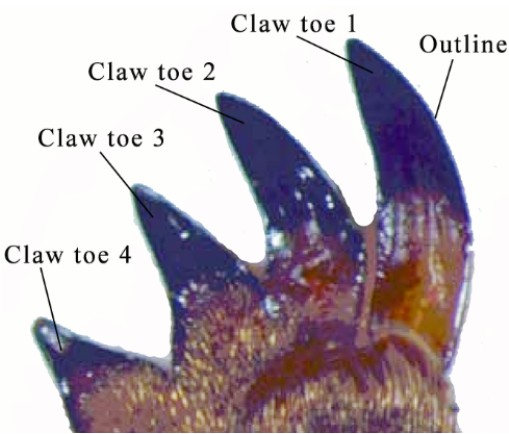

**Figure 4.** Forefoot outline of oriental mole cricket.

The oriental mole cricket has forelegs with four claws, where the outermost claw (claw toe 1 in Figure 4) is well-developed. As such, we used the contour fitting curve of foreleg claw 1 as the bionic prototype [30]. The image analysis tools in Matlab R2018b (MathsWork) were employed to analyze the structure of the oriental mole cricket foreleg claws (Figure 5). In particular, grayscale processing was performed on an image of foreleg claw toe 1, which was captured from Figure 4, followed by binary processing, and the contour of the binary image was then extracted to obtain contour data points. The contour curve was drawn based on these data points.

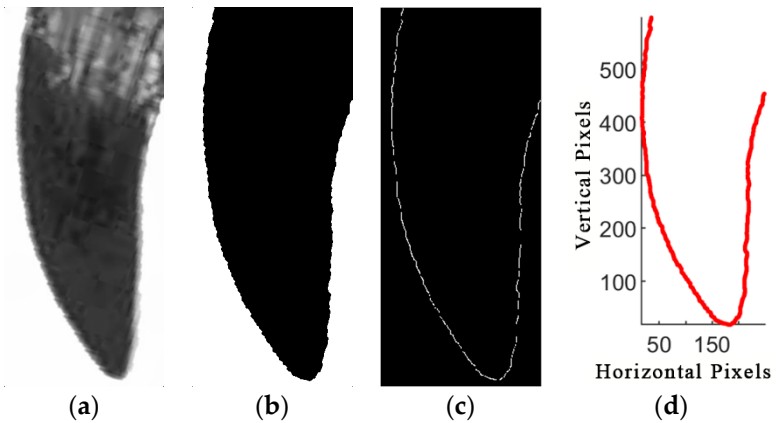

**Figure 5.** Process of extracting contours: (**a**) grayscale image; (**b**) binary image; (**c**) contour extraction; (**d**) contour curve.

The outer contour of the lateral claw toe frequently comes into contact with the soil. The design prototype of the hole-forming device was based on the outer contour. The X and Y two-dimensional coordinate data of the obtained outer contour curve were imported into the Curve Fitting Toolbox of Matlab, and polynomial fitting was performed on the data points. By selecting a third-order polynomial model for fitting, the outer contour of the first claw toe was fitted to a curve (Figure 6).

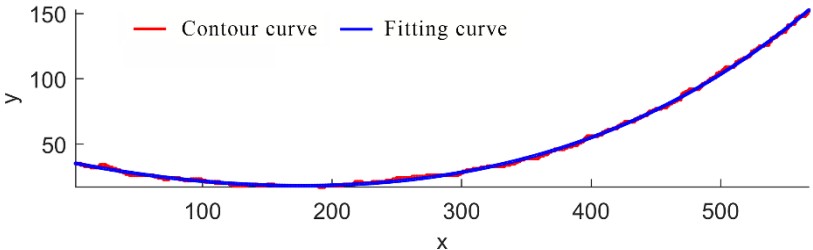

**Figure 6.** External contour fitting curve of claw toe.

The equation for the fitted curve model is given by Formula (8):

$$f(x) = p_1 x^3 + p_2 x^2 + p_3 x + p_4, \qquad (8)$$

where the coefficients of $p_1$, $p_2$, $p_3$, and $p_4$ are $7.144 \times 10^{-7}$, $2.328 \times 10^{-4}$, $-0.1561$, and $34.51$, respectively. The sum of the squares of error (SSE) of the fitting equation was $8.032 \times 10^4$, the coefficient of determination (*R*-square) was 0.9992, and the adjusted coefficient of determination (adjusted *R*-square) was also 0.9992. Furthermore, the root mean square error (RMSE) was 1.077. Thus, the sum of the squares of error of the polynomial fitting model was close to zero, and both the coefficient of determination and the adjusted coefficient of determination were close to 1. This indicated that the model fit the data points well, and the model equation could represent the contour curve of the oriental mole cricket.

The equation for the bionic curve was input into the 3D modeling software Solidworks 2018. Figure 7 presents the resulting design of the three-dimensional model for the bionic hole seeder fixed hole-forming mechanism. The length, width, and thickness of the fixed hole-forming mechanism were 74 mm, 36.3 mm, and 30 mm. The inclination angle was 86°. The structural parameters of the hole-forming mechanism mainly affected the hole size of hole sowing. The length and thickness of the fixed hole-forming mechanism determined the depth and width of the hole.

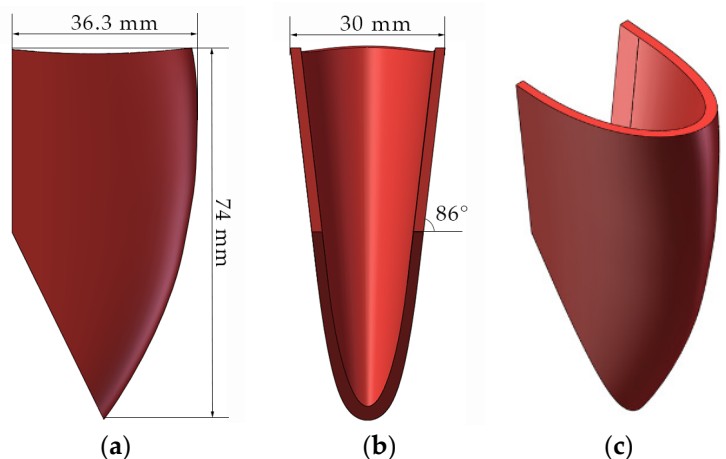

**Figure 7.** Soil contact components of bionic fixed hole-forming device: (**a**) front view; (**b**) lateral view; (**c**) isometric view.

*2.9. Interaction Model between Hole-Forming Device and Soil*

Currently, square and cone-shaped hole-forming devices are commonly used in cotton hole seeders due to their simple structure and manufacturing process and stress condition analysis. In this study, a comparative experiment was conducted between the bionic hole seeder and traditional square and cone-shaped hole-forming devices to eliminate the influence of dimensional parameters on the experimental results. The design dimensions of the three types of hole seeders were essentially consistent. Figure 8 depicts the corresponding three-dimensional models.

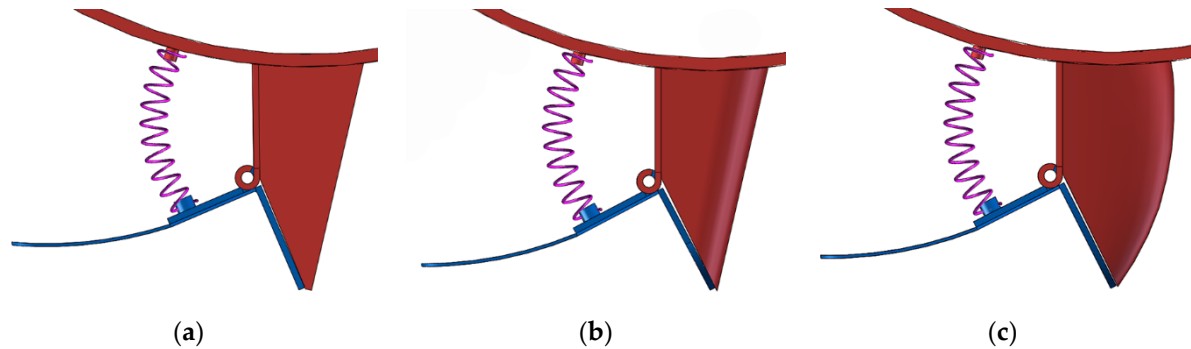

**Figure 8.** Soil contact components of bionic fixed hole-forming device: (**a**) square hole-forming device; (**b**) cone-shaped hole-forming device; and (**c**) bionic hole-forming device.

We adopted the discrete element method and multi-body dynamics (DEM-MBD) method to analyze the interaction mechanism between the main soil-contacting component of the seeder, the hole-forming device, and the soil. First, the hole seeder was modeled in Solidworks, and the three-dimensional model was then imported into the multibody dynamics software Recurdyn V9R2 (FunctionBay). The material properties, connections, forces, motions, and contact parameters were set in the software. The dynamic ring, dynamic disc, and fixed hole-forming device in the model were fixed to each other and rotated along the central axis. The central axis moved forward at a certain speed, and the dynamic hole-forming device rotated about the axis of the fixed hole-forming device with a spring connection between them (Figure 9). After all parameters were set, motion simulations were performed to ensure the correct motion before importing the model into EDEM 2020 (DEM-Solutions).

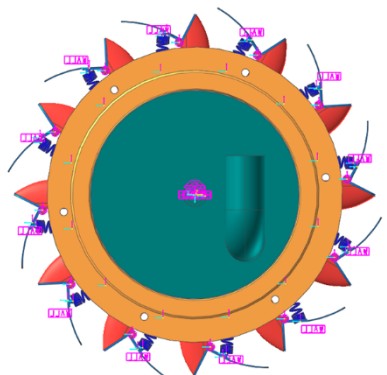

**Figure 9.** RecurDyn simulation model.

In order to facilitate the analysis of the soil disturbance induced by the hole-forming device, the soil in the simulation soil trough was layered according to the main soil layer subjected to the hole-forming device. In particular, the soil was divided into three layers, namely, the upper, middle, and lower layers. The length, width, and height of the simulated soil trough were 600 mm, 150 mm, and 50 mm, respectively, and the contact parameters and size distribution of soil particles in each layer were consistent with the soil parameters calibrated in Section 2.2. The mechanism did not move during the generation of soil particles in the soil trough. The bottom layer of soil was generated first, followed by the middle and upper layers. Approximately 460,000 particles were generated. Figure 10 presents the simulation operating state of the hole-forming device.

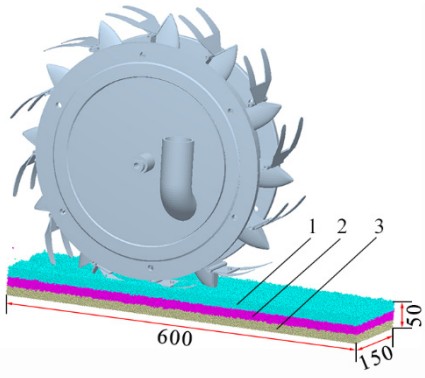

**Figure 10.** Simulation operation status of the hole-forming device: 1—upper soil; 2—middle soil; 3—subsoil.

## 3. Results and Discussion

### 3.1. Stacking Angle Test

Table 8 reports the significance test results of the regression model. The fitting of the regression model was highly significant ($p < 0.01$). $p$-values less than 0.01 were determined for the first-order terms ($X_2$, $X_5$, and $X_7$) of the soil-to-soil static friction coefficient $X_2$, soil-to-steel static friction coefficient $X_5$, and JKR surface energy $X_7$, as well as the second-order term ($X_5^2$) of the soil-to-steel static friction coefficient $X_5$. This indicated that they had a very significant effect on the soil simulation repose angle. Moreover, the first-order term ($X_3$) of the soil-to-soil dynamic friction coefficient $X_3$ had a $p$-value less than 0.05, indicating a significant effect on the soil simulation pile angle. The $p$-value of the lack-of-fit term was 0.1569, which was not significant, indicating that the regression model did not have any other major factors affecting the index. The goodness of fit $R^2$ of the regression equation was 0.9373, revealing a strong fit between the predicted and actual values, and a high degree of explanation between the independent and dependent variable.

**Table 8.** Analysis of variance for regression model.

| Source of Variance | Sum of Squares | Degrees of Freedom | Mean Square Sum | *F*-Value | *p*-Value | Significance |
|---|---|---|---|---|---|---|
| Model | 438.2817 | 14 | 31.3058 | 16.0279 | <0.0001 | ** |
| $X_2$ | 20.6091 | 1 | 20.6091 | 10.5514 | 0.0054 | ** |
| $X_3$ | 16.8002 | 1 | 16.8003 | 8.6014 | 0.0103 | * |
| $X_5$ | 52.9848 | 1 | 52.9848 | 27.1271 | 0.0001 | ** |
| $X_7$ | 319.3022 | 1 | 319.3022 | 163.4758 | <0.0001 | ** |
| $X_2X_3$ | 0.2862 | 1 | 0.2862 | 0.1465 | 0.7072 | |
| $X_2X_5$ | 0.0420 | 1 | 0.0420 | 0.0215 | 0.8853 | |
| $X_2X_7$ | 0.2256 | 1 | 0.2256 | 0.1155 | 0.7387 | |
| $X_3X_5$ | 4.4944 | 1 | 4.4944 | 2.3010 | 0.1501 | |
| $X_3X_7$ | 3.61 | 1 | 3.61 | 1.8482 | 0.1941 | |
| $X_5X_7$ | 0.5776 | 1 | 0.5776 | 0.2957 | 0.5946 | |
| $X_2^2$ | 0.7524 | 1 | 0.7524 | 0.3852 | 0.5441 | |
| $X_3^2$ | 0.9579 | 1 | 0.9579 | 0.4904 | 0.4945 | |
| $X_5^2$ | 18.1350 | 1 | 18.1350 | 9.2847 | 0.0082 | ** |
| $X_7^2$ | 0.0967 | 1 | 0.0967 | 0.0495 | 0.8269 | |
| Residual | 29.2981 | 15 | 1.9532 | | | |
| Lack of fit | 24.4910 | 10 | 2.4491 | 2.5474 | 0.1569 | |
| Pure error | 4.8071 | 5 | 0.9614 | | | |
| Cor total | 467.5798167 | 29 | | | | |

Note: ** denotes an extremely significant impact from the parameter ($p < 0.01$), and * represents a significant impact from the parameter ($p < 0.05$).

Based on the variance analysis results of the regression model, the insignificant factors affecting the soil simulation repose angle in Table 8 were removed. The regression equation for the soil simulation repose angle could be obtained using Formula (9):

$$Y = 41.86 - 0.93X_2 + 0.84X_3 + 1.49X_5 + 3.65X_7 - 0.78X_5^2 \tag{9}$$

With the target of the measured soil repose angle of 39.37°, the soil simulation repose angle regression model was optimized and solved. The optimal solution was obtained when the soil-to-soil static friction coefficient $X_2$ was 0.49, the soil-to-soil dynamic friction coefficient $X_3$ was 0.26, the soil-to-steel static friction coefficient $X_5$ was 0.46, and the JKR surface energy $X_7$ was 0.14 J/m$^2$. Figure 11 depicts the simulation results based on this parameter combination. Comparing the simulation soil pile angle and measured values revealed that they essentially had the same shape.

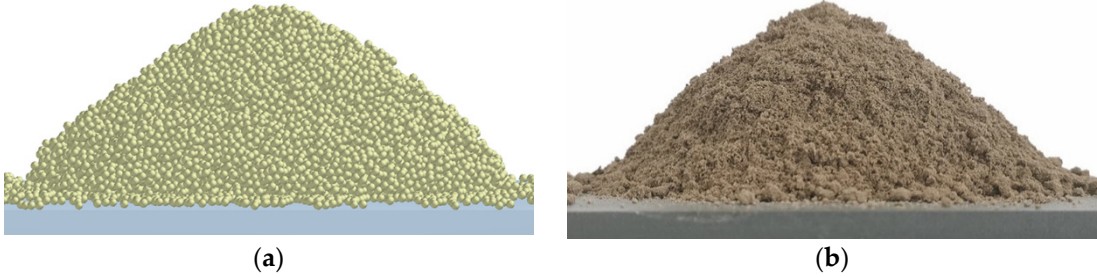

(**a**)      (**b**)

**Figure 11.** Comparison of simulation and measured values of soil stacking angle: (**a**) simulation test; (**b**) actual test.

### 3.2. Hole-Forming Test

In order to analyze the disturbance of different types of hole seeders on the soil, the soil cross-sections at the bottom of the hole following the seeder operation were clipped using the Clipping tool in the EDEM Analyst template. The cross-sections were taken along the *X*-axis for the top view and along the *Z*-axis for the side view (Figures 12 and 13, respectively). The movement range of soil particles in both views caused by the square hole-forming device was larger than that of the other devices, and soil particles from the upper and middle layers were brought into the lower layer. The cone-shaped and bionic hole-forming devices disturbed the soil less, bringing fewer particles from the upper and middle layers into the lower layer. From the top view, it can be seen that the soil disturbance caused by the square and cone-shaped hole-forming devices formed a trapezoidal shape. Moreover, the disturbance range caused by the cone-shaped hole-forming device was smaller than that of the square hole-forming device. The disturbance caused by the bionic hole-forming device formed a triangular shape and created the smallest disturbance range. From the side view, it can be seen that the hole-forming device of the seeder created holes in the soil, and the square and cone-shaped hole-forming devices formed soil ridges behind the holes. However, the bionic hole-forming device hardly formed any soil ridges.

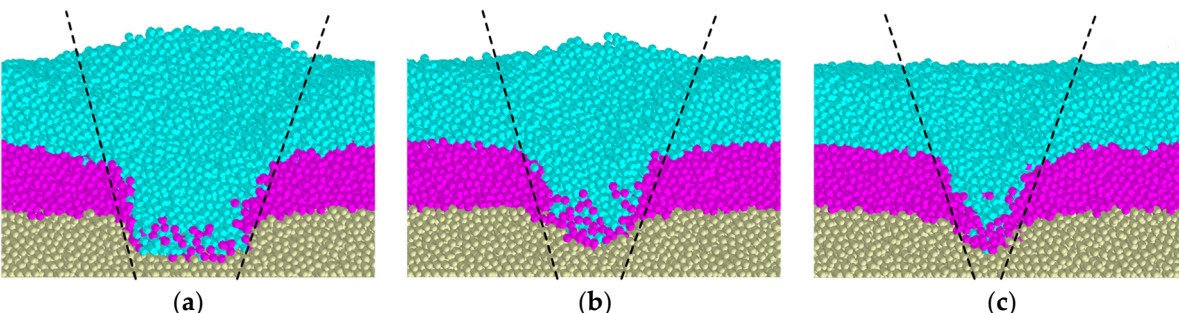

**Figure 12.** Front view of soil particle state after hole-forming operation: (**a**) square hole-forming device; (**b**) cone-shaped hole-forming device; and (**c**) bionic hole-forming device. Blue particles are upper soil, Pink particles are middle soil, yellow particles are subsoil. The regions between dashed lines represent the disturbed soil particles.

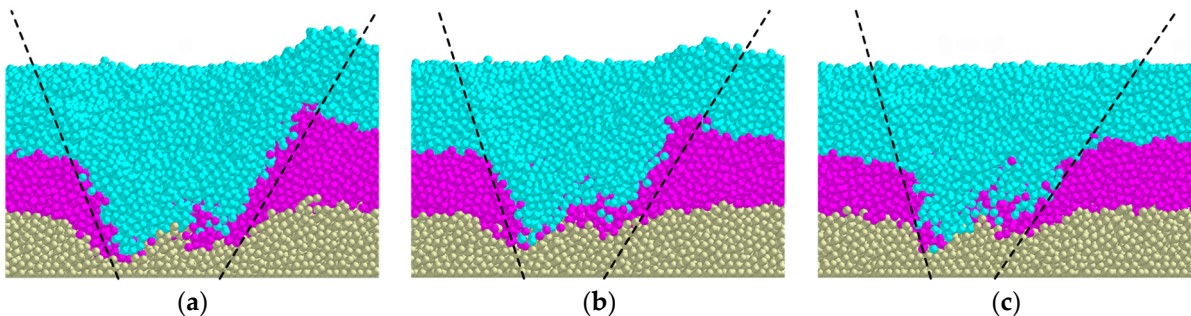

**Figure 13.** Lateral view of soil particle state after hole-forming operation: (**a**) square hole-forming device; (**b**) cone-shaped hole-forming device; and (**c**) bionic hole-forming device. Blue particles are upper soil, pink particles are middle soil, yellow particles are subsoil. The regions between dashed lines represent the disturbed soil particles.

In order to investigate the effect of different types of hole-forming devices on the movement of soil particles, the velocity vector of soil particle groups under the action of different types of hole-forming device was analyzed. Figure 14 presents the velocity vector map of soil particles for hole seeders with different hole-forming devices at 0.44 s. In the figure, the dynamic hole-forming device is about to open, as the soil pressure exerted on it is greater than the pre-tension force of the spring. The maximum velocity of soil particles caused by the square hole-forming device was the highest, reaching 0.386 m/s, followed by the cone-shaped hole-forming device at 0.301 m/s, and the bionic hole-forming device had the lowest velocity at 0.298 m/s. As the contact surface of the square hole-forming device with the soil was a plane, the velocity direction of the soil particles was essentially perpendicular to the contact surface. The lower layers of the soil had a larger linear velocity caused by contact with the hole-forming device, resulting in a larger disturbance to the soil. A similar phenomenon also existed for the cone-shaped hole-forming device, but the disturbance range on the soil was smaller than that of the square hole-forming device. As for the bionic hole-forming device, the contact surface with the soil was a curved contour line, and the velocity direction of the soil particles was along the normal direction of the contour line. This resulted in a more uniform distribution of velocity compared to the other devices. Thus, the simulation results showed that the bionic hole-forming device caused the least disturbance on the soil.

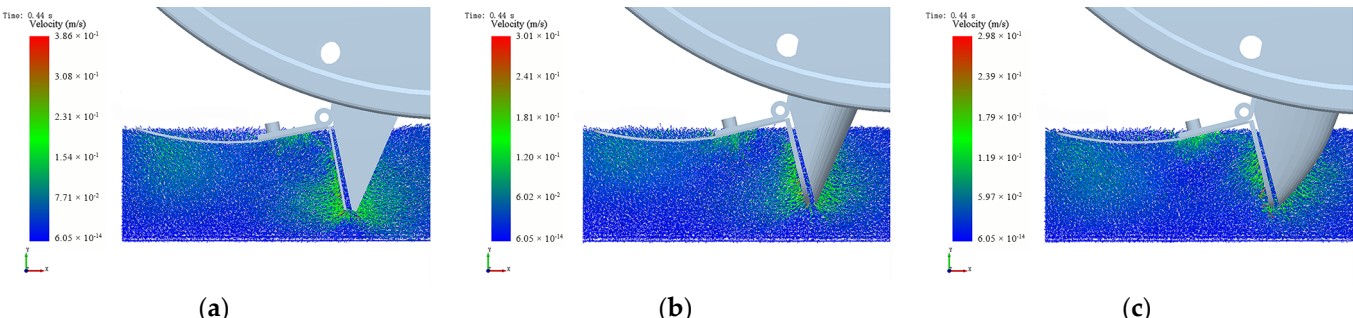

**Figure 14.** Vector plot of soil particle velocity at 0.44 s: (**a**) square hole-forming device; (**b**) cone-shaped hole-forming device; and (**c**) bionic hole-forming device.

Figure 15 presents the velocity vector plot of soil particles for hole seeders with different hole-forming devices at 0.6 s. In the figure, the hole-forming device is beginning to leave the soil as the seeder rotates, and the dynamic hole-forming device starts to close. The maximum velocity of the soil particles caused by the square hole-forming device was the highest, reaching 0.402 m/s, followed by the cone-shaped hole-forming device at 0.349 m/s, and the bionic hole-forming device had the lowest velocity at 0.306 m/s. The movement of the fixed square hole-forming device followed a parabolic trend as the seeder rotated, causing the soil particles to also move backwards in a parabolic trend. As a result, ridges were formed in the soil behind the hole after the operation. The cone-shaped hole-forming device also caused ridges in the soil, yet the ridges induced by the square hole-forming device were larger, as its contact surface with the soil was larger. Although the fixed bionic hole-forming device moved in a parabolic trend, its contact surface with the soil was formed by a curved contour line and did not cause the soil particles to be propelled backwards. As a result, there were barely any soil ridges created behind the hole in the soil. Thus, the simulation results revealed that the bionic hole-forming device caused less disturbance on the soil and did not create any soil ridges.

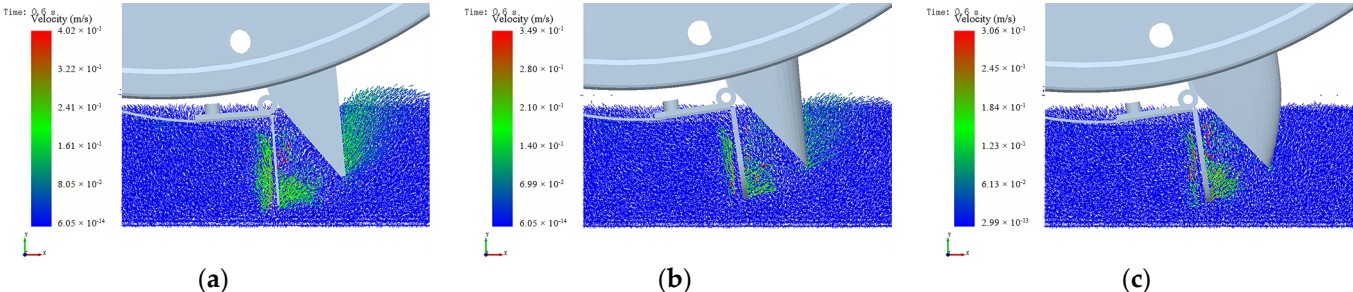

**Figure 15.** Vector plot of soil particle velocity at 0.6 s: (**a**) square hole-forming device; (**b**) cone-shaped hole-forming device; and (**c**) bionic hole-forming device.

Figure 16 presents the maximum force on different hole openers during the hole-forming operation. The force exerted on the hole-forming device gradually increased with the depth of insertion into the soil. When the top of the hole-forming device reached the lowest point in the soil, the force exerted on the hole opener reached its maximum value. Following this, the force exerted on the hole-forming device began to decrease until it completely left the soil. The square hole-forming device experienced more force than the cone-shaped and bionic hole-forming devices, with a maximum force of 9.50 N throughout the entire operation. This could be attributed to the similar upper and lower widths of the square hole-forming device, resulting in a larger contact surface with the soil compared to the cone-shaped and bionic hole-forming devices. This led to a larger force being exerted. Furthermore, the cone-shaped and bionic hole-forming devices had a larger upper width and smaller lower width, resulting in a smaller contact surface area, and hence less force

was exerted. The maximum force exerted on the cone-shaped hole-forming device during the entire operation was 7.96 N, and that on the bionic hole-forming device was 7.51 N. The analysis showed that although the bionic hole-forming device had a larger contact surface with the soil than the cone-shaped hole-forming device, its maximum force during the hole-forming device process was smaller. This indicated a disturbance and resistance reduction effect on the soil during the operation. The soil moisture content is a critical factor that influences the performance of hole-forming devices. An increase in moisture content weakens the cohesion between soil particles, resulting in a reduction in the shear and compressive strength of the soil. Moreover, a high moisture content increases the fluidity of the soil and reduces hole formation performance. On the other hand, a low moisture content raises the resistance faced during the sowing process. Therefore, planting under optimal soil conditions is advantageous for enhancing the quality of sowing.

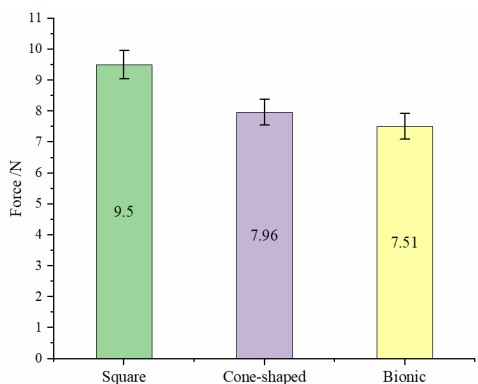

**Figure 16.** Maximum stress of different hole-forming devices.

## 4. Conclusions

A typical soil discrete element particle model was established, and a bionic hole-forming device was designed. The hole-forming processes and performance of various hole-forming devices were investigated by DEM-MBD simulation tests. The mechanism of motion interaction between the soil and hole-forming device was analyzed, and the following conclusions could be drawn:

(1)　The moisture content of the soil in the cotton field during the appropriate sowing period was 14.46%, the density was 1.37 g/cm$^3$, and the bulk density was 1547.7 Pa. Gravel (particle diameter > 1–2 mm); sand (particle diameter 0.075–1 mm); and silt (particle diameter < 0.075 mm) accounted for 34.78%, 52.12%, and 13.1%, respectively, of the particle size distribution of the soil. The natural rest angle of the soil was 39.34°.

(2)　Based on physical experiments and simulation experiments, the combined Hertz–Mindlin and JKR contact model was used to establish a second-order regression model for the soil accumulation angle and soil contact parameters through optimized experimental methods. The target optimization was based on the measured soil accumulation angle of 39.37°. The optimal solution was obtained when the static friction coefficient between soil and soil was 0.49, the dynamic friction coefficient between soil and soil was 0.26, the static friction coefficient between soil and steel was 0.46, and the surface energy of JKR was 0.14 J/m$^2$.

(3)　A bionic hole-forming device was designed based on bionic technology using the oriental mole cricket as a prototype. A discrete element method (DEM) and multi-body dynamics (MBD) coupled algorithm was used to establish a discrete element simulation model of the hole-forming device and soil. A hole-formation experiment was carried out using an EDEM–Recurdyn joint simulation. The interaction between different structural forms of the hole-forming device and soil during the hole-formation process was analyzed. The simulation results showed that the disturbance of the soil caused by the bionic hole-forming device was small, and it did not cause soil ridges

to form. Moreover, the soil resistance of the hole-forming device was the smallest, at 7.51 N.

This work should be helpful in improving hole-forming performance and reducing soil resistance during the cotton planting process. However, there was still some discrepancy between the results of the simulation experiment and the actual field operation. Future work will explore the performance of hole-forming devices and the interaction mechanism between hole-forming devices and soil under different soil conditions in field experiments.

**Author Contributions:** Conceptualization, L.W. and J.X.; methodology, L.W. and J.X.; software, L.W.; validation, L.W., X.W. and J.X.; formal analysis, X.L.; investigation, W.G. and X.H.; data curation, S.H.; writing—original draft preparation, L.W.; writing—review and editing, L.W. and J.X.; project administration, L.W.; funding acquisition, L.W. All authors have read and agreed to the published version of the manuscript.

**Funding:** This research was supported by the Bintuan Science and Technology Program (2022CB001-06 and 2023AB005) and the President Fund from Tarim University (TDZKYS202302).

**Institutional Review Board Statement:** Not applicable.

**Data Availability Statement:** The data presented in this study are available upon request from the authors.

**Acknowledgments:** The authors would like to thank their schools and colleges, as well as the funding providers of the project. All support and assistance are sincerely appreciated.

**Conflicts of Interest:** The authors declare no conflict of interest.

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
