# Peer review of "Study on the Mechanism of Motion Interaction between Soil and a Bionic Hole-Forming Device"

_agriculture, doi:10.3390/agriculture13071421_

Round 1

Reviewer 1 Report

In this paper, a simulated hole-forming device was designed, and a simulation model was established using the discrete element method and multi-body dynamics to simulate the performance of the simulated hole-forming device and the conventional device. This is a very meaningful work, but the research content of this paper still has the following problems:

(1)Soil moisture content has an important influence on the performance parameters of the hole-forming device. The discrete element model used in this paper is a fixed moisture content, and it is suggested that the performance of the hole-forming at different moisture contents be discussed.

(2)It is recommended that the content of parts 3.1, 3.2 and 3.3 be transferred to part 2 Materials and Methods, as the determination of the discrete element model parameters is not part of the results.

(3)Suggest supplementing the structural parameters of the hole forming device and their impact on performance.

Formatting and grammar enhancements are recommended.

Author Response

(1)Soil moisture content has an important influence on the performance parameters of the hole-forming device. The discrete element model used in this paper is a fixed moisture content, and it is suggested that the performance of the hole-forming at different moisture contents be discussed.

Response: Thank you for your kind advice. We have supplied the performance of the hole-forming at different moisture contents. “The soil moisture content is a critical factor that influences the performance of the hole-forming device. An increase in moisture content weakens the cohesion between soil particles, resulting in a reduction in the shear and compressive strength of the soil. Moreover, high moisture content increases the fluidity of the soil, reduces hole formation performance. On the other hand, low moisture content raises the resistance faced during the sowing process. Therefore, planting under optimal soil conditions is advantageous for enhancing the quality of sowing.” (Line 442-449)

(2)It is recommended that the content of parts 3.1, 3.2 and 3.3 be transferred to part 2 Materials and Methods, as the determination of the discrete element model parameters is not part of the results.

Response: Thank you for your kind advice. We have transferred “3.1. Stacking Angle Test” and “3.2. Plackett-Burman Test” to part 2 Materials and Methods, and merged into “2.7. Stacking Angle Test”.

(3)Suggest supplementing the structural parameters of the hole forming device and their impact on performance.

Response: Thank you for your kind advice. We have supplemented the structural parameters of the hole forming device and their impact on performance. “The length, width and thickness of fixed hole-forming mechanism are 74 mm, 36.3mm and 30mm. The inclination angle is 86°. The structural parameters of the hole forming mechanism main effect the hole size of hole sowing. The length and thickness of fixed hole-forming mechanism determine the depth and width of the hole.” (Line 292-296)

Reviewer 2 Report

In the manuscript " agriculture-2494040 " entitled Study on the Mechanism of Motion Interaction between Soil and a Bionic Hole-Forming Device, a bionic hole-forming device was designed based on the oriental mole cricket, a typical soil discrete element particle model was established. The simulation experiment was conducted to compare the performance of the bionic hole-forming device with conventional hole-forming devices using discrete element method and multi-body dynamics (DEM-MBD) coupled simulations. The mechanism of motion interaction between soil and a bionic hole-forming device was revealed.

Technically the manuscript is well written and its content is relevant, unpublished and opportune. I believe the study will be of interest to Agriculture readers.

I only have a few suggests for the authors to improve this paper:

1. Include quantifiable results to back this statement up in the abstract.

2. Include the aims/objectives of your study in the introduction.

3. Line 91: “...... during the suitable sowing period”.  Should specify when this period was.

4. Explain the units of all variables in the formula.

5. Line 210: “...... an image of foreleg claw toe 1.” Specify how this image was obtained.

6. “3.1. Stacking Angle Test” seems to be more suited in the methods section?

7. In the conclusion should include: (1) Did your results fulfil the study aims? (2) What are the limitations of your study? (3) Applications of this work.

Minor editing of English language required

Author Response

  1. Include quantifiable results to back this statement up in the abstract.

Response: Thank you for your kind advice. We have add quantifiable results to back this statement up in the abstract. “Compared to traditional square and cone-shaped hole-forming devices, the soil resistance of the bionic hole-forming device was the smallest , at 7.51 N.” (Line 22-24)

  1. Include the aims/objectives of your study in the introduction.

Response: Thank you for your kind advice. We have add the aims of this study in the introduction. “The purpose of this study is to: (1) perform discrete element method and multi-body dynamics (DEM-MBD) modeling to analyze hole-forming processes and performances under various hole-forming devices, (2) provide optimization approaches for improvement of hole-forming devices.”(Line 81-84)

  1. Line 91: “...... during the suitable sowing period”.  Should specify when this period was.

Response: Thanks for your careful review. We have specified the period.” In order to determine the model parameters, the samples were collected using the five-point sampling method from the plowed and harrowed cotton field soil during the suitable sowing period (March to April).” (Line 89-91)

  1. Explain the units of all variables in the formula.

Response: Thanks for your careful review. We have explained the units of all variables in the formula

  1. Line 210: “...... an image of foreleg claw toe 1.” Specify how this image was obtained.

Response: Thanks for your careful review. We have specified how this image was obtained. “which was captured from Figure 4” (Line 268)

  1.  “3.1. Stacking Angle Test” seems to be more suited in the methods section?

Response: Thank you for your kind advice. We have transferred “3.1. Stacking Angle Test” to “2.7. Stacking Angle Test”

  1. In the conclusion should include: (1) Did your results fulfil the study aims? (2) What are the limitations of your study? (3) Applications of this work.

Response: Thank you for your kind advice. We have added some comment in the conclusion. “The typical soil discrete element particle model was established and a bionic hole-forming device was designed. The hole-forming processes and performances of various hole-forming devices were investigated by DEM-MBD simulation tests. The mechanism of motion interaction between soil and hole-forming device was analyzed, and the conclusions can be drawn as follows:” (Line 441-445). “This work should be helpful in improving hole-forming performance and reducing the soil resistance during the cotton planting process. However, there is still some dis-crepancy between the results of the simulation experiment and the actual field operation. Future work will explore hole-forming performance of hole-forming device and interaction mechanism between hole-forming device and soil under different soil conditions in field experiment.” (Line 468-473)

Reviewer 3 Report

The paper has clear ideas, sufficient experiments, and detailed content. It introduced some interesting results. However, some problems need to be revised. The detail comments are explained below.

1)        Is it confirmed in Table 1 that it is 0.075 mm instead of 0.75 mm?

2)        The 3D model in Figure 6 is not very consistent with Figure 4.

3)        In line 270, if the soil particle size is modeled according to 2.2 and includes particles of 0.075mm, then the soil model must have more than 460000 particles.

4)        The validation test in Figure 11 should be different from the previous screening test method, and it is recommended to use penetration test.

5)        Why are the pictures in Figure 12 and Figure 13 identical?

The English expression should be modified to make it more academic.

Author Response

1) Is it confirmed in Table 1 that it is 0.075 mm instead of 0.75 mm?

Response: Thank you for your kind advice. We have confirmed in Table 1 that the size range of sand is 0.075 mm to 1 mm. Reference is “Song, S.L.; Tang, Z.H.; Zheng, X.; Liu, J.B.; Meng, X.J.; Liang, Y.C. Calibration of  the discrete element parameters for the soil model of cotton field after plowing in Xinjiang of China. Transact. Chin Soc. Agric Eng. 2021, 37, 63-70.” We have added the reference in manuscript.

2)  The 3D model in Figure 6 is not very consistent with Figure 4.

Response: Thanks for your careful review. The oriental mole cricket has forelegs with four claws, where the outermost claw (claw toe 1 in Figure 4) is well-developed and has the most interaction with soil. As such, we just used the contour fitting curve of the outline of foreleg claw 1 as the bionic prototype, to design the hole-forming device. So only the contour curve of 3D model in figure 6 is very consistent with outline of claw toe 1 in Figure 4.

3)  In line 270, if the soil particle size is modeled according to 2.2 and includes particles of 0.075mm, then the soil model must have more than 460000 particles.

Response: Thanks for your careful review. Due to the soil particle size of 0.075 mm is too small, and have less impact on the operation of the hole-forming device, the soil particle sizes don’t include particles of 0.075mm. We have specified the soil particles size. “According to the soil particle size distribution experiment, the soil particle size is mainly concentrated in 1 mm, so the soil particle radius was set as 1 mm in the simulation.” (Line 208-210)

4)  The validation test in Figure 11 should be different from the previous screening test method, and it is recommended to use penetration test.

Response: Thank you for your kind advice. The screening test method is a common method of calibrating soil discrete model, which is used to determine the optimal combination of simulation parameters and verify its effectiveness. This method has also been applied in multiple soil discrete element modeling studies, such as:

  • Fang, W.Q.; Wang, X.Z.; Han, D.L.; Chen, X.G. Review of Material Parameter Calibration Method. Agriculture 2022, 12(5), 706; https://dorg/10.3390/agriculture12050706
  • Shi, L.; Zhao, W.; Sun, W. Parameter calibration of soil particles contact model of farmland soil in northwest arid region based on discrete element method. Trans. Chin. Soc. Agric. Eng. 2017, 33, 181–187.
  • Wang, X.L.; Hu H.; Wang, Q.J.; Li, H.W.; He, J.; Chen, W.Z. Calibration method of soil contact characteristic parameters based on DEM theory. Chin. Soc. Agric. Mach. 2017, 48, 78–85.
